∂ | **Open Peer Review** | Biotechnology | Research Article

# Comparison of CRISPR-Cas9, CRISPR-Cas12f1, and CRISPR-Cas3 in eradicating resistance genes KPC-2 and IMP-4

Jun Huang,[1,2] Kanghui Ding,[3] Jiahui Chen,[4] Jiao Fan,[4] Luyao Huang,[4] Shaofu Qiu,[1] Ligui Wang,[1] Xinying Du,[1] Chao Wang,[1] Haifeng Pan,[3] Zhengquan Yuan,[1] Hongbo Liu,[1] Hongbin Song[1]

**ABSTRACT** Bacterial plasmid encoding antibiotic resistance could be eradicated by various CRISPR systems, such as CRISPR-Cas9, Cas12f1, and Cas3. However, the efficacy of these gene editing tools against bacterial resistance has not been systematically assessed and compared. This study eliminates carbapenem resistance genes KPC-2 and IMP-4 via CRISPR-Cas9, Cas12f1, and Cas3 systems, respectively. The eradication efficiency of the three CRISPR systems was evaluated. First, the target sites for the three CRISPR systems were designed within the regions 542–576 bp of the KPC-2 gene and 213–248 bp of the IMP-4 gene, respectively. The recombinant CRISPR plasmids were transformed into *Escherichia coli* carrying KPC-2 or IMP-4-encoding plasmid. Colony PCR of transformants showed that KPC-2 and IMP-4 were eradicated by the three different CRISPR systems, and the elimination efficacy was both 100.00%. The drug sensitivity test results showed that the resistant *E. coli* strain was resensitized to ampicillin. In addition, the three CRISPR plasmids could block the horizontal transfer of drug-resistant plasmids, with a blocking rate as high as 99%. Importantly, a qPCR assay was performed to analyze the copy number changes of drug-resistant plasmids in *E. coli* cells. The results indicated that CRISPR-Cas3 showed higher eradication efficiency than CRISPR-Cas9 and Cas12f1 systems.

**IMPORTANCE** With the continuous development and application of CRISPR-based resistance removal technologies, CRISPR-Cas9, Cas12f1, and Cas3 have gradually come into focus. However, it remains uncertain which system exhibits more potent efficacy in the removal of bacterial resistance. This study verifies that CRISPR-Cas9, Cas12f1, and Cas3 can eradicate the carbapenem-resistant genes KPC-2 and IMP-4 and restore the sensitivity of drug-resistant model bacteria to antibiotics. Among the three CRISPR systems, the CRISPR-Cas3 system showed the highest eradication efficiency. Although each system has its advantages and characteristics, our results provide guidance on the selection of the CRISPR system from the perspective of resistance gene removal efficiency, contributing to the further application of CRISPR-based bacterial resistance removal technologies.

**KEYWORDS** antimicrobial resistance, KPC-2, IMP-4, CRISPR-Cas9, CRISPR-Cas12f1, CRISPR-Cas3

**Peer Reviewer** Sharmi Naha, ICMR - National Institute for Research in Bacterial Infections, Kolkata, West Bengal, India

Address correspondence to Haifeng Pan, panhaifeng@ahmu.edu.cn, Zhengquan Yuan, yzhengquan66@163.com, Hongbo Liu, mailoflhb@126.com, or Hongbin Song, hongbinsong@263.net.

The authors declare no conflict of interest.

See the funding table on p. 12.

Over the past two decades, antibiotic resistance has emerged as a significant global threat to public health and clinical medicine (1, 2). The misuse of antibiotics in both human medicine and animal husbandry has facilitated the development of antibiotic resistance genes (ARGs) (3). These ARGs can be found on bacterial chromosomes as well as on mobile genetic elements, such as plasmids (4). Plasmids, in particular, serve as crucial vectors for ARGs, thereby promoting their dissemination among bacterial populations (5). The acquisition of plasmid-mediated ARGs is the primary driver of

bacterial resistance dissemination (6). Carbapenem antibiotics, which are broad-spectrum β-lactamase bactericidal agents, are crucial for the clinical treatment of severe infections. Their stability against β-lactamases underscores their significance as essential antibiotics for human anti-infection therapies (7). Resistance to carbapenems in *Klebsiella pneumoniae* is primarily due to the production of KPC-type and IMP-type carbapenemases. The genes encoding these enzymes, KPC-2 and IMP-4, are typically located on conjugative plasmids, which can be transmitted both within and between bacterial species (8).

Therefore, it is imperative to develop new antibacterial technologies to eliminate drug-resistant genes and prevent the spread of drug-resistant plasmids among bacteria. In most bacteria, the absence of non-homologous end-joining mechanisms means that DNA double-strand breaks mediated by Cas proteins cannot be spontaneously repaired (9). The CRISPR-Cas system can specifically identify and cleave drug resistance gene DNA, demonstrating significant potential in blocking and controlling the spread of bacterial resistance (10). The CRISPR-Cas9 system, in particular, has been widely applied in combating bacterial resistance (11, 12). However, the large size and insufficient efficiency of CRISPR-Cas9 have been persistent drawbacks in the application of this system (13). With the continuous development and refinement of CRISPR technology, more CRISPR-based antibacterial techniques, such as CRISPR-Cas3 and CRISPR-Cas12f1, are emerging for the eradication of bacterial resistance. CRISPR-Cas3 can degrade target DNA processively and is used to rapidly generate large deletions in bacterial genomes (14). A notable feature of CRISPR-Cas12f1 is its small size—only half that of CRISPR-Cas9—making it easier to deliver to organisms (15). Given the current prevalence of carbapenem resistance genes KPC-2 and IMP-4, there is an urgent need to develop novel technologies to combat these antibiotic resistance genes. To date, there are no reports on the use of CRISPR-Cas12f1 and CRISPR-Cas3 to eradicate the KPC-2 and IMP-4 resistance genes, and it remains unclear which system is more efficient in removing these resistance genes.

In this study, we designed targets for three CRISPR systems within the regions 542–576 bp of the KPC-2 gene and 213–248 bp of the IMP-4 gene, respectively. We verified that CRISPR-Cas9, CRISPR-Cas12f1, and CRISPR-Cas3 can eliminate the drug resistance genes KPC-2 and IMP-4. Following the eradication of these drug-resistant plasmids, we observed the restoration of antibiotic sensitivity in model drug-resistant bacteria. We assessed the efficiency of the three CRISPR systems using quantitative PCR (qPCR). The qPCR results indicated that the CRISPR-Cas3 system exhibited the highest eradication efficiency among the three systems. Therefore, the CRISPR-Cas3 system holds significant potential as a novel technology for preventing and controlling bacterial resistance. It is expected to address the issue of bacterial resistance and restore the sensitivity of bacteria to antibiotics, offering substantial application value in the field of combating drug-resistant bacteria.

## MATERIALS AND METHODS

### Bacterial strains, plasmids, and growth conditions

All strains and plasmids used or constructed in this study are listed in Table S1. *Escherichia coli* DH5α was used as chemically competent cells for plasmid transformation. The plasmid pCas9 was a gift from Luciano Marraffini (Addgene plasmid #42876; http://n2t.net/addgene:42876; RRID: Addgene_42876) (16). The plasmid pCas3cRh was a gift from Joseph Bondy-Denomy (Addgene plasmid #133773; http://n2t.net/addgene:133773; RRID: Addgene_133773) (14). The plasmid pCas12f1 was obtained from previous research (15). *E. coli* was grown in Luria-Bertani (LB; 2 g yeast extract, 4 g NaCl, and 4 g tryptone per 400 mL) broth or on LB agar (LB supplemented with 6 g agar per 400 mL) plates at 37°C. When necessary, the following antibiotics were supplemented at the specified concentrations: tetracycline (10 mg/mL), chloramphenicol (50 mg/mL), gentamicin (15 mg/mL), and kanamycin (50 mg/mL).

## Target design

According to the CRISPR-Cas9 PAM sequence (protospacer adjacent motif sequence) principles, we selected a 30-nucleotide sequence upstream of the NGG motif in the multidrug resistance gene sequence as the spacer. For CRISPR-Cas12f1, a 20-nucleotide sequence upstream of the TTTN motif was selected as the spacer. For CRISPR-Cas3, we selected the antisense strand of a 34-nucleotide sequence upstream of the GAA motif as the spacer. All spacers used in this study are listed in Table S2. The targets for the three CRISPR systems were designed within the regions 542–576 bp of the KPC-2 gene and 213–248 bp of the IMP-4 gene, respectively, to enable a comparative analysis of their eradication efficiency.

## Plasmid construction

All oligonucleotides used in this study are listed in Table S2. For pCas9, oligonucleotides were synthesized with a length of 30 nucleotides targeting the sgRNA sequence. A sticky end AAAC or G was added at the beginning or end of the forward oligonucleotide, while AAAAC was added at the beginning of the reverse oligonucleotide. For pCas12f1, oligonucleotides were synthesized with a length of 20 nucleotides targeting the sgRNA sequence. A sticky end GAAC was added at the beginning of the forward oligonucleotide, while GGCC was added at the beginning of the reverse oligonucleotide. For pCas3, oligonucleotides were synthesized with a length of 34 nucleotides targeting the sgRNA sequence. A sticky end GAAAC or G was added at the beginning or end of the forward oligonucleotide, while GCGAC or G was added at the beginning or end of the reverse oligonucleotide. Next, pCas9, pCas3, and pCas12f1 were digested with the restriction endonuclease BsaI, and the digested products were ligated with the corresponding annealed fragments using a rapid ligation kit. Finally, we successfully obtained CRISPR plasmids targeting KPC-2, comprising three pCas9 plasmids, three pCas12f1 plasmids, and three pCas3 plasmids, as well as plasmids targeting IMP-4. To obtain the drug resistance model plasmids, fragments of KPC-2 and IMP-4 were retrieved from NCBI and submitted to GenBank under accession numbers MG764553 and MF344566, respectively (17, 18). The KPC-2 fragment was amplified using primers KPC-2 (KpnI)-F and KPC-2 (SalI)-R, and the IMP-4 fragment was amplified using primers IMP-4 (KpnI)-F and IMP-4 (SalI)-R. Following amplification, these fragments were digested with KpnI and SalI and then ligated into the corresponding sites in pSEVA551, yielding the plasmids pKPC-2 and pIMP-4. The model drug-resistant plasmids were then transformed into *E. coli* DH5α to obtain the model drug-resistant bacteria.

## Preparation and transformation of competent cells

*E. coli* DH5α containing the resistant plasmid pKPC-2 or pIMP-4 was prepared into competent cells using the TransEasy ultra-efficient competent cell preparation kit (GeneCopoeia). Single colonies were picked and inoculated into LB liquid medium containing 10 µg/mL tetracycline. The cultures were placed in a shaker at 37°C with shaking at 180–220 rpm until the $OD_{600}$ reached approximately 0.5. The cultures were then subjected to an ice bath for 10 minutes, followed by centrifugation to discard the supernatant. The cell pellet was resuspended first in 50 µL of Solution A (GeneCopoeia) and then in 50 µL of Solution B (GeneCopoeia), mixed well by pipetting, and kept on ice for an additional 10 minutes. This completed the preparation of the competent cells.

The prepared competent cells were mixed with 1,000 ng of CRISPR plasmid and kept on ice for 30 minutes. Following this, a heat shock was performed at 42°C for 90 seconds, after which the cells were immediately placed back on ice for 3 minutes. The transformants were then inoculated in 1 mL of SOC medium and incubated at 37°C with shaking at 180–220 rpm for 1 hour. The transformants were screened on LB agar plates containing the corresponding antibiotics and incubated overnight at 37°C. An empty CRISPR plasmid was used as a negative control.

## PCR and qPCR experiments

After transformation, single colonies were randomly selected to evaluate the elimination efficacy. The presence of the CRISPR plasmid was detected by PCR using primers pCas9-F/R, pCas12f1-F/R, or pCas3-F/R. To screen for colonies with the drug resistance gene deletion, PCR was performed using primers KPC-2-F/R or IMP-4-F/R. Drug-resistant strains transformed with the empty vector CRISPR plasmid served as positive controls. All primers used in this study are listed in Table S2.

To further analyze the elimination efficacy of the three CRISPR systems on the drug resistance genes KPC-2 or IMP-4, the relative fold change of pKPC-2 or pIMP-4 at 12, 24, and 48 hours was determined by qPCR. Bacterial genomic nucleic acids were extracted using the MagaBio Pathogen DNA/RNA Purification Kit (Bioer, Hangzhou, China). The qPCR reactions were performed using the probe qPCR mix (TaKaRa, Beijing, China) with specific primers and probes for the KPC-2 and IMP-4 genes, as well as for the chromosomal 16S ribosomal RNA gene. All reactions were repeated five times. The $2^{-\Delta\Delta CT}$ method was used to calculate the relative fold change of pKPC-2 or pIMP-4 in the experimental group compared to the blank control group.

## Drug sensitivity test

The changes in drug sensitivity phenotype were detected using the drug sensitivity E-test strip (Liofilchem, Italy) and the Phoenix drug sensitivity plate. According to the guidelines of the Institute for Clinical and Laboratory Standards (11), the minimum inhibitory concentration (MIC) values of ampicillin were compared between the control group and the experimental group.

## Statistical analysis

All statistical analyses and graphs were generated using GraphPad Prism version 9.0 (GraphPad Software Inc., San Diego, CA, USA). A Student's $t$-test was used to calculate the $P$-values. The bars in the graphs represent the average values of three biological replicates, and the error bars indicate the standard deviation. All $P$-values reported in the article are less than 0.05. The significance levels are denoted as follows: *$P < 0.05$, **$P < 0.01$, ***$P < 0.001$, and ****$P < 0.0001$.

## RESULTS

### Validation of the effect of three CRISPR systems in eradicating drug resistance genes KPC-2 and IMP-4

A total of 52 CRISPR-Cas9 targets, 9 CRISPR-Cas12f1 targets, and 9 CRISPR-Cas3 targets were designed within the KPC-2 resistance genes. Similarly, 37 CRISPR-Cas9 targets, 19 CRISPR-Cas12f1 targets, and 16 CRISPR-Cas3 targets were identified within the IMP-4 resistance genes. The crRNA coding sequence of pCas9 was replaced with a 30-nucleotide oligonucleotide corresponding to the sgRNA sequence, generating pCas9-k, pCas9-k1, pCas9-k2, pCas9-i, pCas9-i1, and pCas9-i2, respectively (Fig. S1A). The crRNA coding sequence of pCas12f1 was replaced with a 20-nucleotide oligonucleotide corresponding to the sgRNA sequence, generating pCas12f1-k, pCas12f1-k1, pCas12f1-k2, pCas12f1-i, pCas12f1-i1, and pCas12f1-i2, respectively (Fig. S1B). The crRNA coding sequence of pCas3 was replaced with a 34-nucleotide oligonucleotide corresponding to the sgRNA sequence, generating pCas3-k, pCas3-k1, pCas3-k2, pCas3-i, pCas3-i1, and pCas3-i2, respectively (Fig. S1C). Additionally, to construct model resistant plasmids containing the KPC-2 or IMP-4 genes, plasmid pSEVA551 was selected as the backbone.

To assess the efficacy of three CRISPR systems in mediating plasmid elimination in *E. coli*, five single colonies were randomly selected from both the experimental and control groups. The results demonstrated that all five colonies in the experimental group were negative for KPC-2 or IMP-4, indicating that the KPC-2 or IMP-4 genes could not be amplified by PCR. This suggests a 100% elimination efficacy of the three CRISPR

systems in the five randomly selected transformants. In contrast, in the control group, the Cas9, Cas12f1, or Cas3 nucleases lacked the guidance of sgRNA and, therefore, did not specifically clear KPC-2 or IMP-4. The experimental results are summarized in Fig. 1, with additional details and extended data provided in Fig. S2. The bacterial colony PCR results confirmed that the three specific CRISPR systems effectively eradicated the drug resistance genes KPC-2 and IMP-4, demonstrating 100% efficiency at each of the three CRISPR systems' target sites.

To further evaluate the performance of these systems, we selected pCas9-k, pCas12f1-k, and pCas3-k targeting KPC-2, as well as pCas9-i, pCas12f1-i, and pCas3-i targeting IMP-4, which are located adjacent to each other, to compare the eradication efficiency of the three CRISPR systems (Fig. S3).

## Changes in bacterial susceptibility phenotype

The threat posed by bacteria producing KPC-2 and IMP-4 resistance genes lies in their resistance to β-lactam antibiotics. The Phoenix susceptibility plate test was employed to determine bacterial sensitivity to various antibiotics. As shown in Tables S3 to S5, following the elimination of the KPC-2 resistance gene by the three CRISPR systems, the sensitivity of the experimental group to amoxicillin-clavulanic acid, ampicillin, ampicillin-sulbactam, aztreonam, piperacillin, and piperacillin-tazobactam was restored, in contrast to the control group, which remained resistant to these antibiotics. Similarly, as presented in Tables S6 to S8, after the elimination of the IMP-4 resistance gene by the three CRISPR systems, the experimental group showed restored sensitivity to amoxicillin-clavulanic acid, ampicillin, cefotaxime, and ceftazidime, while the control group continued to exhibit resistance to these antibiotics.

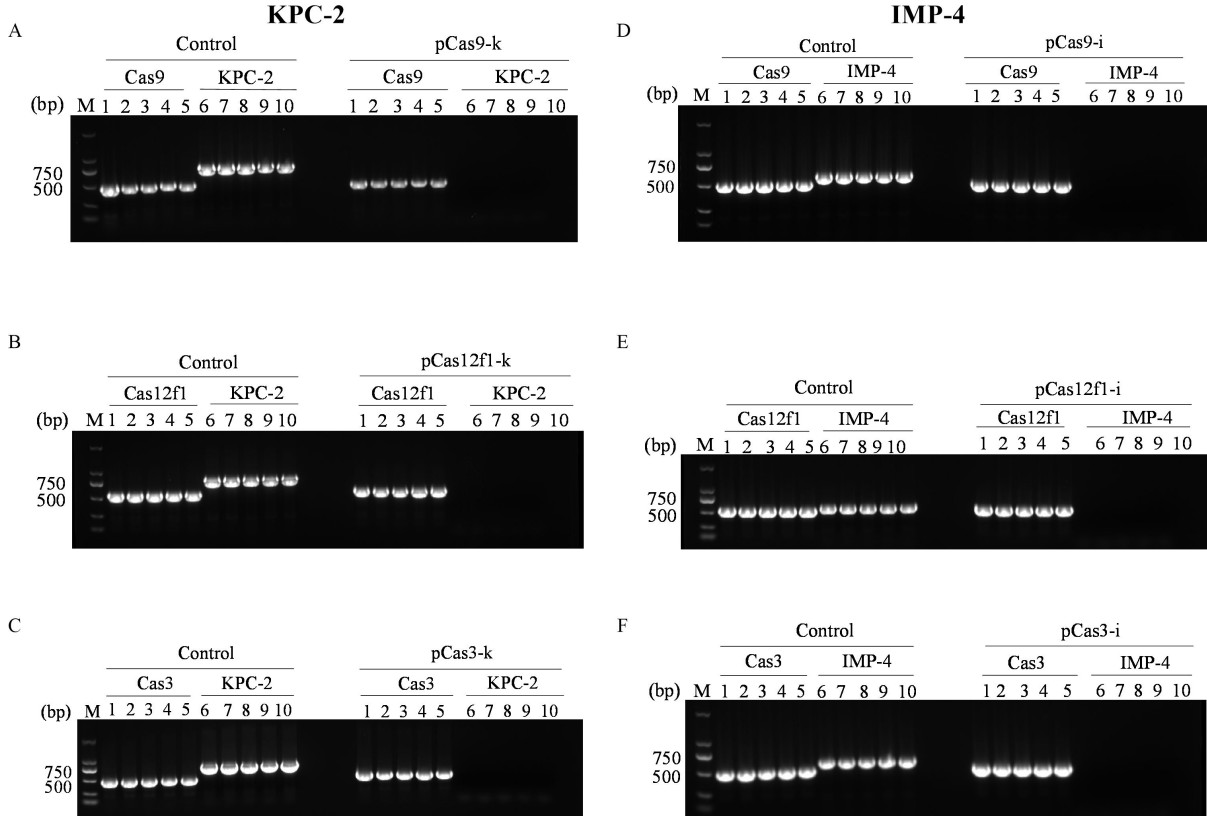

FIG 1 Confirmation of KPC-2/IMP-4 gene presence in *E. coli* DH5α + pKPC-2/pIMP-4 by PCR amplification. Control as a negative control is transformed into an empty vector plasmid. Lane M represents 2,000 bp DNA molecular markers. KPC-2 was eliminated by CRISPR-Cas9 (A), CRISPR-Cas12f1 (B), and CRISPR-Cas3 (C). IMP-4 was eliminated by CRISPR-Cas9 (D), CRISPR-Cas12f1 (E), and CRISPR-Cas3 (F).

E-test susceptibility test strips were used to detect changes in the MIC values of ampicillin in the experimental group. As shown in Fig. 2A through C, compared with the control group, the MIC value of the bacteria in the experimental group decreased approximately 170.67-fold after KPC-2 was specifically eliminated by the three CRISPR systems ($n = 3$, unpaired $t$-test, $P < 0.0001$). The MIC value in the control group was approximately 256 µg/mL, while the MIC value in the experimental group dropped to about 1.5 µg/mL. Similarly, as shown in Fig. 2D through F, the MIC value of the bacteria in the experimental group decreased approximately 21.3-fold after IMP-4 was specifically eliminated by the three CRISPR systems ($n = 3$, unpaired $t$-test, $P < 0.0001$). The MIC value

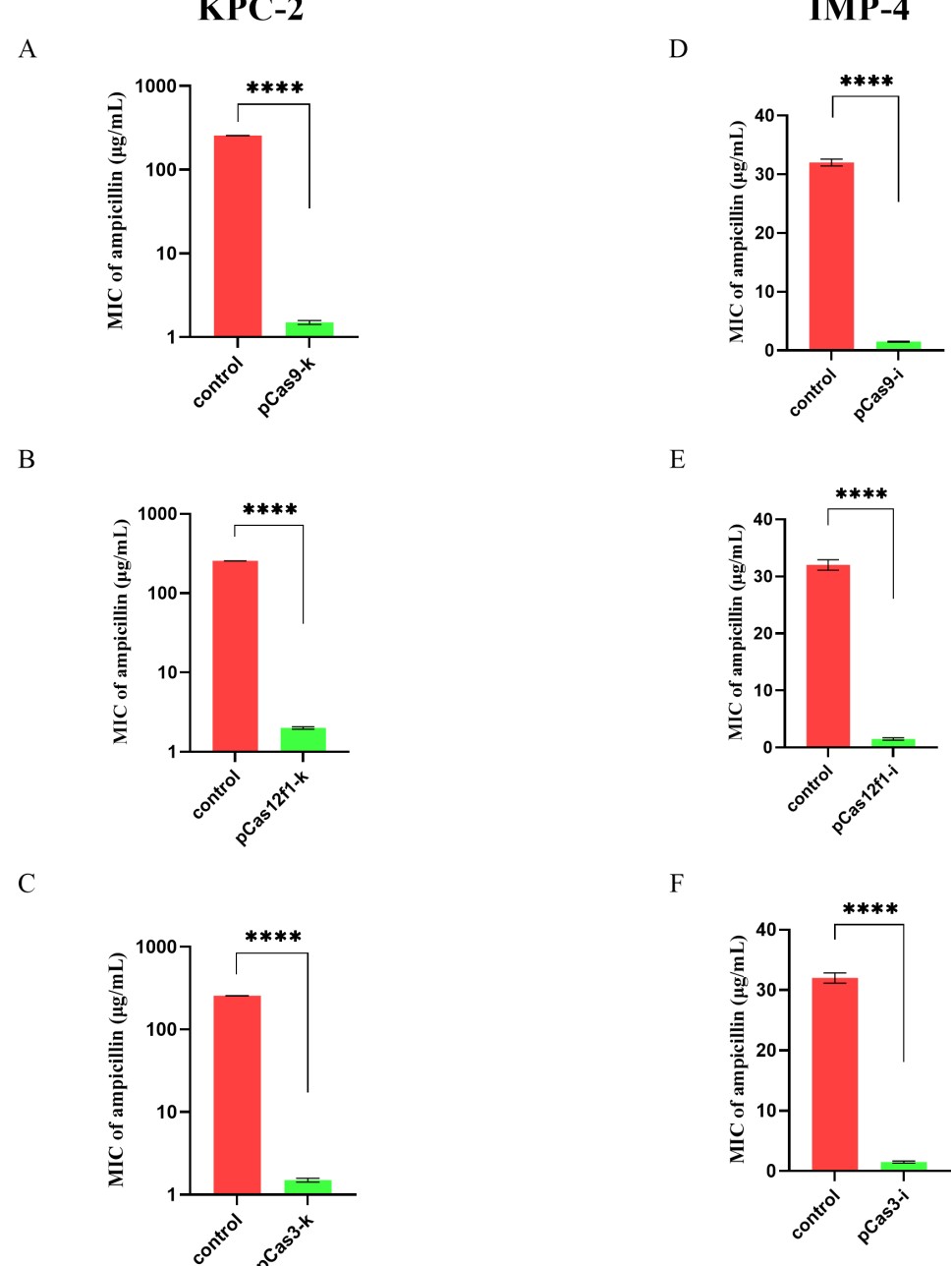

**FIG 2** KPC-2/IMP-4-mediated drug resistance elimination by three CRISPR systems. KPC-2-mediated drug resistance elimination by CRISPR-Cas9 (A), CRISPR-Cas12f1 (B), CRISPR-Cas3 (C), and statistical analysis of ampicillin MIC value (A–C). IMP-4-mediated drug resistance elimination by CRISPR-Cas9 (D), CRISPR-Cas12f1 (E), CRISPR-Cas3 (F), and statistical analysis of ampicillin MIC value (D–F).

in the control group was about 32 µg/mL, whereas the MIC value in the experimental group decreased to approximately 1.5 µg/mL.

## Three CRISPR systems can block the acquisition of drug-resistant plasmids by bacteria through transformation

This part of the experiment analyzes whether the three CRISPR systems can block the spread of drug-resistant plasmids through chemical transformation. The resistant plasmid pKPC-2 was transformed into both the experimental and control groups via chemical transformation. As shown in Fig. 3A, the DH5α strain containing plasmid pCas9-k reduced the number of transformants by 3,106.6 times compared to the control plasmid pCas9 ($1.55 \times 10^4$ vs 5 CFU/mL; $P = 0.0012$). In Fig. 3B, the DH5α strain containing plasmid pCas12f1-k reduced the number of transformants by 1,030.2 times compared to the control plasmid pCas12f1 ($1.13 \times 10^4$ vs 11 CFU/mL; $P = 0.0005$). Figure 3C shows that the DH5α strain containing plasmid pCas3-k reduced the number of transformants by 2,083.3 times compared to the control plasmid pCas3 ($3.33 \times 10^4$ vs 16 CFU/mL; $P = 0.0012$).

Similarly, the resistant plasmid pIMP-4 was transformed into both the experimental and control groups. As shown in Fig. 3D, the DH5α strain containing plasmid pCas9-i reduced the number of transformants by 4,285.7 times compared to the control plasmid pCas9 ($3 \times 10^4$ vs 7 CFU/mL; $P = 0.04$). In Fig. 3E, the DH5α strain containing plasmid pCas12f1-i reduced the number of transformants by 5,880 times compared to the control plasmid pCas12f1 ($2.94 \times 10^4$ vs 5 CFU/mL; $P = 0.0009$). Figure 3F shows that the DH5α strain containing plasmid pCas3-i reduced the number of transformants by 20,222 times compared to the control plasmid pCas3 ($6.06 \times 10^4$ vs 3 CFU/mL; $P = 0.0005$).

These results demonstrate that the three CRISPR systems can effectively block the acquisition of drug-resistant plasmids by bacteria through transformation, achieving a blocking efficiency of 99%.

## CRISPR system assists antibiotics in killing drug-resistant bacteria

As shown in Fig. 4A, the KPC-2 gene was targeted by three CRISPR systems at intervals of 12, 24, and 48 hours. After each time point, the bacterial solution from each experimental group was spread onto LB plates containing ampicillin and cultured overnight. The CRISPR-Cas3 system demonstrated the ability to assist ampicillin in killing resistant bacteria, with activity initiating at 12 hours and persisting until 48 hours. In contrast, the CRISPR-Cas12f1 system exhibited activity starting at 24 hours and continued to function until 48 hours, whereas the CRISPR-Cas9 system required a full 48 hours to effectively assist ampicillin in eliminating resistant bacteria.

As shown in Fig. 4B, the IMP-4 gene was targeted by the three CRISPR systems at the same intervals of 12, 24, and 48 hours. The bacterial solution from each experimental group was spread onto LB plates containing ampicillin and cultured overnight. The CRISPR-Cas3 system demonstrated the ability to assist ampicillin in eliminating resistant bacteria, with activity initiating at 12 hours and persisting until 48 hours. In contrast, both the CRISPR-Cas12f1 and CRISPR-Cas9 systems exhibited activity only after 24 hours. These experimental results indicate that, compared to the CRISPR-Cas9 and CRISPR-Cas12f1 systems, the CRISPR-Cas3 system can assist antibiotics in exerting bactericidal efficacy in the shortest amount of time.

## Analysis of the elimination efficacy of three CRISPR systems on KPC-2 and IMP-4

To more accurately evaluate elimination efficacy, the relative fold change of plasmids at 12, 24, and 48 hours was determined by qPCR. As shown in Fig. 5A, significant differences in plasmid relative fold change were observed between pCas9-k, pCas12f1-k, and pCas3-k. The elimination efficiency of pCas3-k was greater than 99.99% at the 12th hour and continued to be this effective until the 48th hour. In comparison, the elimination

**KPC-2**
**IMP-4**

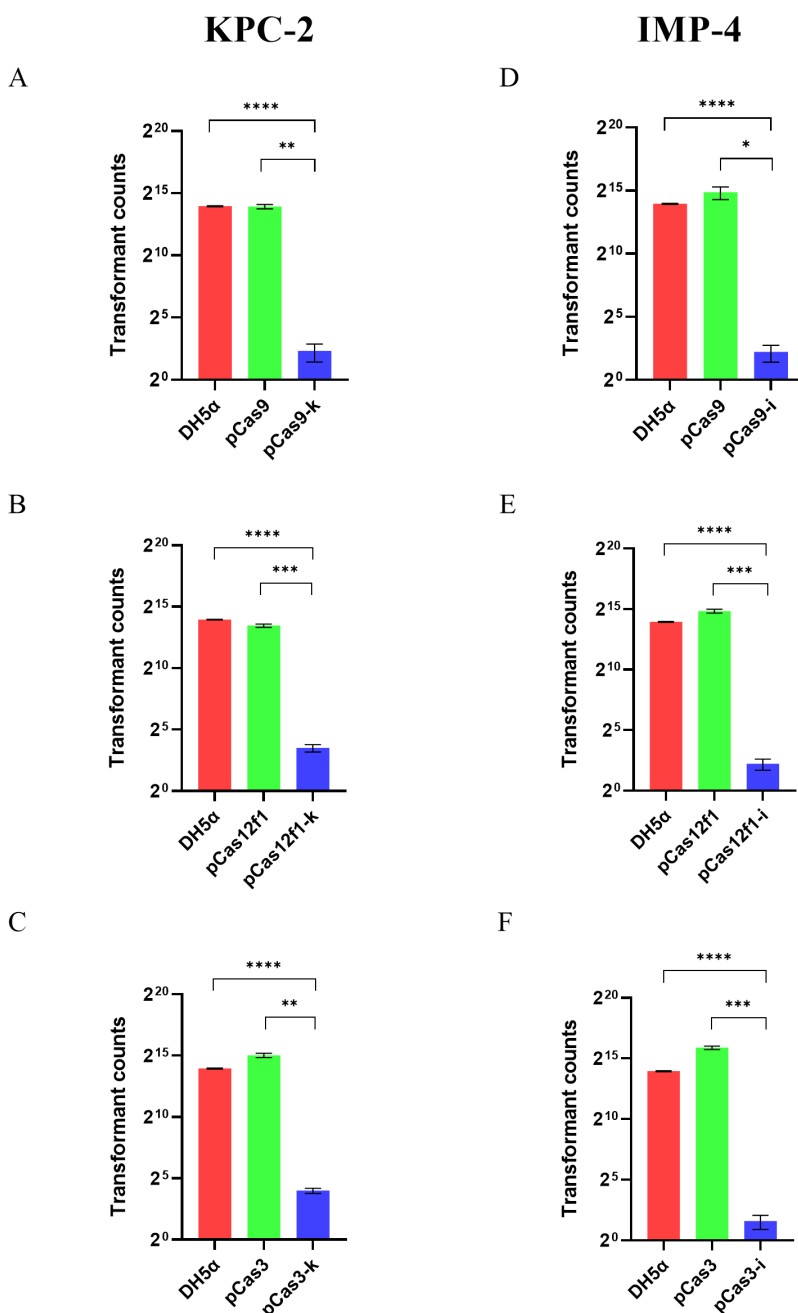

**FIG 3** Three CRISPR systems can block the uptake of drug-resistant plasmid pKPC-2 or pIMP-4 by bacteria through chemical transformation. The pKPC-2 was transformed into the blank control DH5α, pCas9 and pCas9-k (A), pCas12f1 and pCas12f1-k (B), and pCas3 and pCas3-k (C), respectively. The pIMP-4 was transformed into the blank control DH5α, pCas9 and pCas9-i (D), pCas12f1 and pCas12f1-i (E), and pCas3 and pCas3-i (F), respectively. The number of transformants was counted.

efficiency of pCas12f1-k reached 99.99% at the 24th hour, while pCas9-k achieved 99.99% elimination efficiency at the 48th hour. This clearly indicates that the CRISPR-Cas3 system can effectively eliminate the KPC-2 gene in the shortest time.

Similarly, as shown in Fig. 5B, significant differences in plasmid relative fold change were observed between pCas9-i, pCas12f1-i, and pCas3-i. The elimination efficiency of pCas3-i was greater than 99.99% at the 12th hour and remained this effective until the 48th hour. In contrast, both pCas12f1-i and pCas9-i achieved 99.99% elimination

A

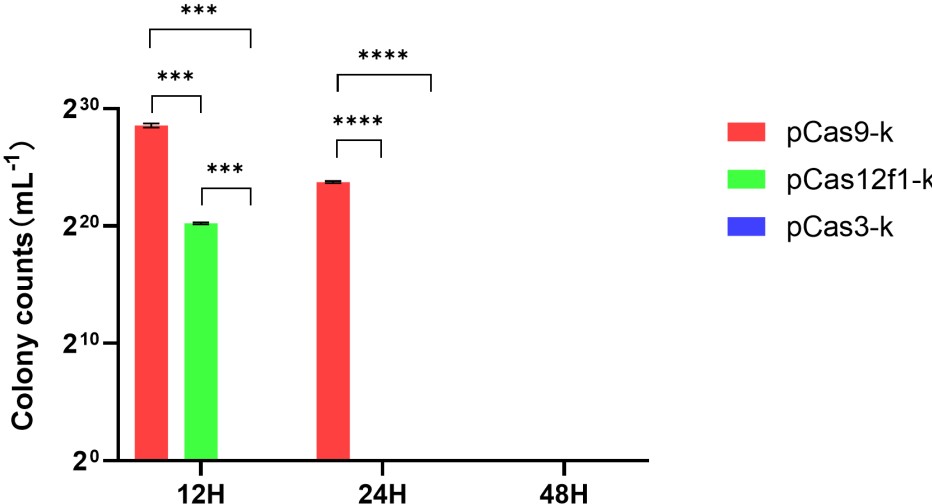

B

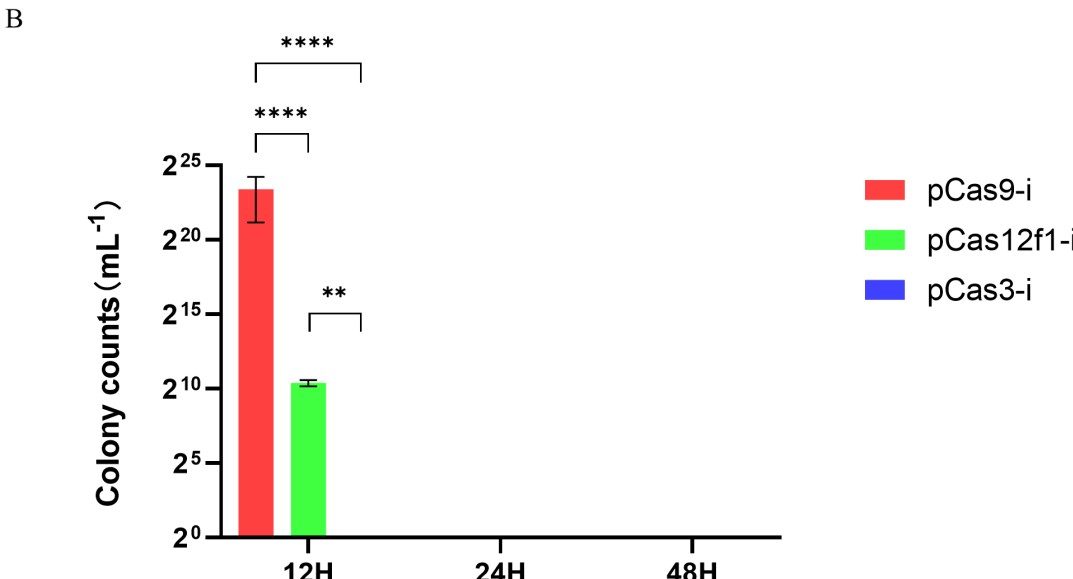

**FIG 4** Three CRISPR systems can assist ampicillin to kill drug-resistant bacteria. (A) The bactericidal efficacy of ampicillin assisted by three CRISPR systems in the removal of drug resistance gene KPC-2 at 12, 24, and 48 hours. (B) The bactericidal efficacy of ampicillin assisted by three CRISPR systems in the removal of drug resistance gene IMP-4 at 12, 24, and 48 hours.

efficiency by the 24th hour. This demonstrates that the CRISPR-Cas3 system can effectively eliminate the IMP-4 gene in the shortest time.

## DISCUSSION

Antibiotic resistance poses a serious threat to global public health (19). The emergence and spread of antibiotic resistance genes such as KPC-2 and IMP-4 present significant challenges to the treatment of clinical infections. Plasmids serve as reservoirs for horizontal gene transfer and are important carriers for the dissemination of drug resistance genes (3). Therefore, the elimination of drug-resistant plasmids can be an effective strategy to slow the spread of antibiotic resistance in bacteria. The CRISPR-Cas system is an adaptive immune system in bacteria that protects against the invasion of

A

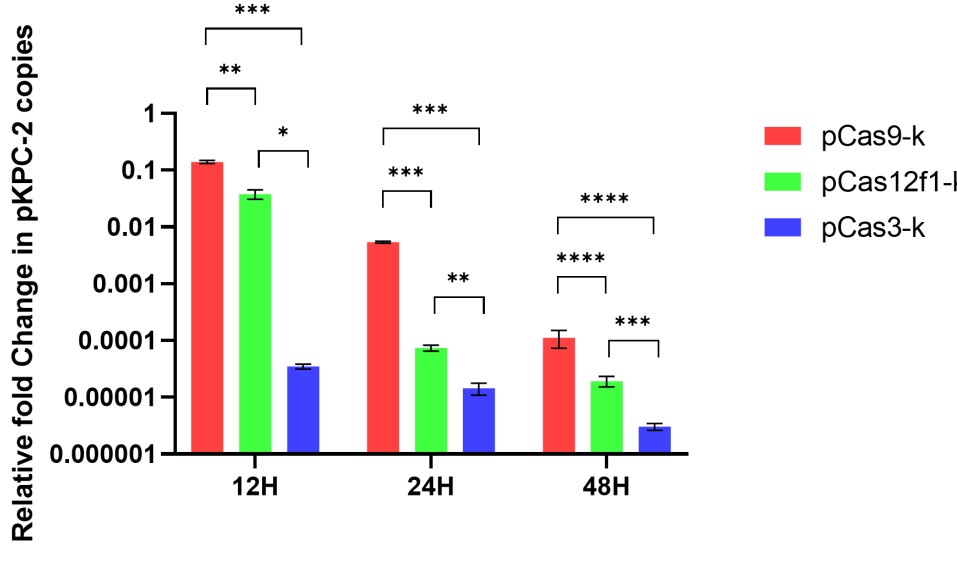

B

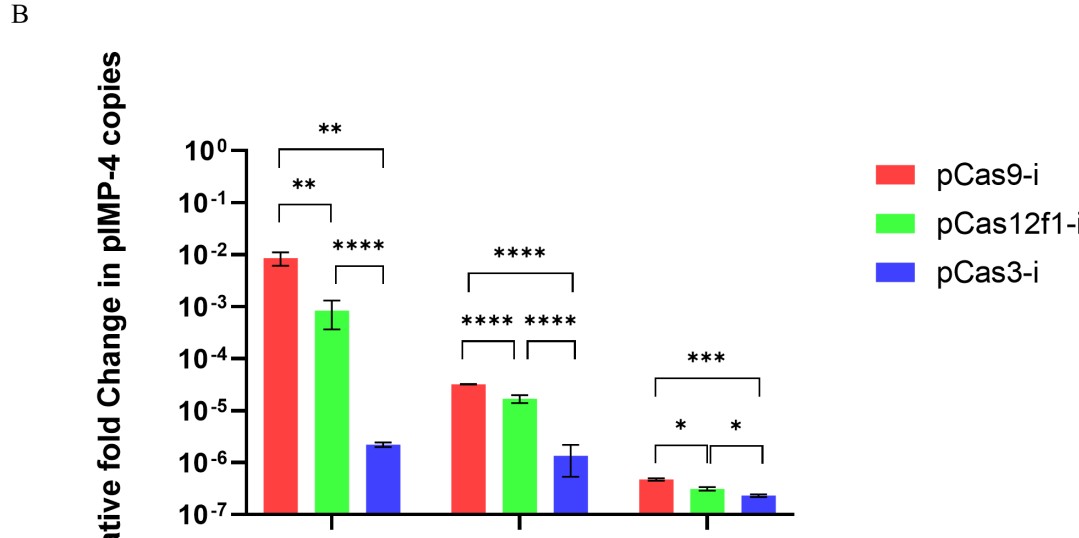

FIG 5 The relative fold change of plasmid pKPC-2 or pIMP-4. (A) The relative fold change of plasmid pSEVA551-KPC−2 at 12, 24, and 48 hours. (B) The relative fold change of plasmid pIMP-4 at 12, 24, and 48 hours.

foreign nucleic acids (20). This system's characteristics provide an effective strategy for eradicating bacterial drug-resistant plasmids and restoring bacterial sensitivity to antibiotics (11). In this study, we developed an efficient strategy to combat resistant bacteria by combining CRISPR-Cas3-based sensitization with conventional antibiotic treatment. This approach restored the efficacy of low-cost drugs and effectively eradicated resistant *E. coli* cells without the emergence of new resistance against CRISPR-Cas3.

CRISPR-Cas9, CRISPR-Cas12f1, and CRISPR-Cas3 were investigated for their ability to eradicate the resistance genes KPC-2 and IMP-4, thereby restoring bacterial sensitivity to antibiotics. Each of the three CRISPR systems was transformed into drug-resistant model bacteria carrying the resistant plasmids pKPC-2 and pIMP-4, respectively. Bacterial colony PCR results demonstrated that the resistance genes KPC-2 and IMP-4 were successfully eliminated by all three CRISPR systems. Subsequent drug sensitivity tests confirmed that the elimination of the drug-resistant plasmids resulted in the recovery of

antibiotic sensitivity in the model bacteria. These findings also indicated that the three CRISPR systems can prevent bacteria from acquiring drug-resistant plasmids through transformation. The time required for CRISPR plasmid transformation to form transformants allows for the CRISPR systems to eradicate the resistance genes, resulting in no significant difference in the removal efficiency among the three CRISPR systems, as shown by colony PCR and the prevention of drug resistance spread.

The main differences among the three CRISPR systems—CRISPR-Cas9, CRISPR-Cas12f1, and CRISPR-Cas3—lie in their PAM requirements and eradication efficiencies. The PAM design of CRISPR-Cas9 is more flexible compared to CRISPR-Cas12f1 and CRISPR-Cas3, allowing CRISPR-Cas9 to target a broader range of sequences. This study explored the differences in the ability of CRISPR-Cas9, CRISPR-Cas12f1, and CRISPR-Cas3 to remove resistance genes. To quantitatively analyze the elimination efficiency of these CRISPR systems, three time points—12, 24, and 48 hours—were selected to measure the eradication of the resistance genes KPC-2 and IMP-4. It was observed that CRISPR-Cas3 exhibited exceptionally high elimination efficiency for both KPC-2 and IMP-4. Additionally, the experimental results indicated that the CRISPR-Cas3 system can assist antibiotics in restoring bactericidal efficiency in the shortest time. This suggests that the CRISPR-Cas3 system holds significant application value in the field of bacterial resistance elimination.

The structural differences and cleavage mechanisms of the Cas proteins may account for the variations in clearance efficiency among the three CRISPR systems. The CRISPR-Cas9 system employs a single DNA endonuclease, Cas9, which cleaves each DNA strand using distinct nuclease domains (HNH or RuvC) (21). The resulting double-strand breaks are then repaired by either error-prone non-homologous end joining (22) or high-fidelity homology-directed repair (23). The V-type nuclease (Cas12) contains a RuvC-like nuclease domain that cleaves both DNA strands, whereas the type II nuclease (Cas9) includes an HNH nuclease domain in addition to the RuvC-like structure (13). Each domain of Cas9 is responsible for digesting a single strand of the target DNA (13). Un1Cas12f1 exhibits high genome editing efficiency due to its ability to form asymmetric homodimers, its low cytotoxicity, and its smaller gene size, which leads to increased Cas protein production (13, 24). The Cas12f1 molecule, consisting of 529 amino acids, is significantly smaller than the commonly used SpCas9 (1,368 amino acids) (25), making CRISPR-Cas12f1 easier to deliver into organisms. In comparison, CRISPR-Cas3 demonstrated high elimination efficiency for drug resistance genes. The CRISPR-Cas3 system generates a single-stranded gap at the DNA sequence, followed by continuous exonuclease degradation of the target strand. In contrast, CRISPR-Cas9 only induces DNA double-strand breaks without additional degradation (26). Furthermore, CRISPR-Cas3 can enhance DNA cleavage efficiency by interfering with DNA repair mechanisms (26). Different CRISPR systems exhibit distinct PAM locations, with the targets of CRISPR-Cas9 and CRISPR-Cas3 systems positioned upstream of the PAM and those of CRISPR-Cas12f1 positioned downstream. These variations lead to differences in sgRNA structure, such as changes in GC content, dimer formation, and specificity, which may ultimately affect the efficiency of the CRISPR system (27). In addition, the sgRNA-Cas complex searches for a suitable target site stochastically through three-dimensional diffusion, sampling multiple target sites for complementarity (28, 29). Bergman et al. (30) propose that densely packed regions may overload the CRISPR machinery with numerous non-complementary target sites, necessitating the sampling of additional sites and reducing the likelihood of encountering a suitable one, thereby compromising the efficiency of the CRISPR system. Finally, the off-target effects of the CRISPR system also significantly impact its targeting efficiency (31).

Given that spontaneous plasmid transformation rarely occurs in natural environments, the autonomous diffusion of CRISPR systems between bacteria remains a challenge for their clinical application. Future studies should focus on developing more efficient vectors for delivering CRISPR systems.

## Conclusion

This study proposes strategies to eliminate bacterial resistance using the CRISPR-Cas9, CRISPR-Cas12f1, and CRISPR-Cas3 systems. These CRISPR systems were employed to target the bacterial resistance genes KPC-2 and IMP-4, block the spread of drug resistance genes among bacteria, and reduce the overall abundance of drug resistance. The highly efficient CRISPR-Cas3 system was identified, showing significant potential to be developed into a new technology for preventing and controlling bacterial resistance. This system is expected to address the problem of bacterial resistance and restore bacterial sensitivity to antibiotics, offering substantial application value in combating drug-resistant bacteria.

## ACKNOWLEDGMENTS

This work was supported by grants from the National Science Foundation of China (32141003 and 32300080) and the National Science and Technology Major Project of China (2021YFC2301102).

## AUTHOR AFFILIATIONS

[1]Department of Infectious Disease Prevention and Control, Chinese People's Liberation Army Center for Disease Control and Prevention, Beijing, China
[2]Department of Human Anatomy and Histology, School of Basic Medicine, Capital Medical University, Beijing, China
[3]Department of Epidemiology and Biostatistics, School of Public Health, Anhui Medical University, Hefei, China
[4]Department of Epidemiology, School of Public Health, China Medical University, Shenyang, China

## AUTHOR ORCIDs

Jun Huang  http://orcid.org/0009-0006-1827-5696
Haifeng Pan  http://orcid.org/0000-0001-8218-5747
Hongbo Liu  http://orcid.org/0000-0002-2320-3682
Hongbin Song  http://orcid.org/0000-0002-2781-3031

## FUNDING

| Funder | Grant(s) | Author(s) |
| --- | --- | --- |
| National Science Foundation of China | 32141003 | Hongbin Song |
| National Science Foundation of China | 32300080 | Hongbo Liu |
| National Science and Technology Major Project | 2021YFC2301102 | Hongbo Liu |

## AUTHOR CONTRIBUTIONS

Jun Huang, Data curation, Formal analysis, Investigation, Writing – original draft | Kanghui Ding, Data curation | Jiahui Chen, Data curation | Jiao Fan, Data curation | Luyao Huang, Data curation | Shaofu Qiu, Data curation | Ligui Wang, Data curation | Xinying Du, Data curation | Chao Wang, Data curation | Zhengquan Yuan, Supervision, Writing – review and editing | Hongbo Liu, Funding acquisition, Supervision, Writing – review and editing | Hongbin Song, Funding acquisition, Project administration, Resources, Supervision, Writing – review and editing.

## DATA AVAILABILITY

The authors confirm that the data supporting the findings of this study are available within the article and its supplemental material.

## ADDITIONAL FILES

The following material is available online.

### Supplemental Material

**Supplemental figures and tables (Spectrum02572-24-s0001.docx).** Figures S1 to S3 and Tables S1 to S8.

### Open Peer Review

**PEER REVIEW HISTORY (review-history.pdf).** An accounting of the reviewer comments and feedback.

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
