## [Reviewer comments · Microbiology Spectrum]

Microbiology Spectrum

Comparison of CRISPR-Cas9, CRISPR-Cas12f1 and CRISPR-Cas3 in eradicating resistance genes KPC-2 and IMP-4

Jun Huang, Kanghui Ding, Jiahui Chen, Jiao Fan, Luyao Huang, Shaofu Qiu, Ligui Wang, Xinying Du, Chao Wang, Haifeng Pan, Zhengquan Yuan, Hongbo Liu, and Hongbin Song

Corresponding Author(s): Hongbin Song, Chinese PLA Center for Disease Control and Prevention

Review Timeline:

Submission Date:	October 10, 2024
Editorial Decision:	November 22, 2024
Revision Received:	January 23, 2025
Accepted:	February 22, 2025

Editor: Silvia Cardona

Reviewer(s): Disclosure of reviewer identity is with reference to reviewer comments included in decision letter(s). The following individuals involved in review of your submission have agreed to reveal their identity: Sharmi Naha (Reviewer #2)

Transaction Report:

DOI: <https://doi.org/10.1128/spectrum.02572-24>

Re: Spectrum02572-24 (Comparison of CRISPR-Cas9, CRISPR-Cas12f1 and CRISPR-Cas3 in eradicating resistance genes KPC-2 and IMP-4)

Dear Prof. Hongbin Song:

Thank you for submitting your manuscript to Microbiology Spectrum. Your article has been reviewed by two experts in the field. Both reviewers found the findings interesting and have provide comments that, if addressed, will improve the quality of the work, substantially.

Their recommendations are provided below.

Revision Guidelines

Sincerely,
Silvia Cardona
Editor
Microbiology Spectrum

Reviewer #1 (Comments for the Author):

The manuscript by Huang et al. compared the efficiency of three CRISPR-Cas systems-CRISPR-Cas9, CRISPR-Cas12f1, and CRISPR-Cas3-in targeting antibiotic resistance genes KPC-2 and IMP-4 in E. coli. The authors demonstrated that all three systems achieved 100% eradication efficiency. Furthermore, they identified CRISPR-Cas3 as particularly effective in assisting

antibiotics to kill drug-resistant bacteria.

Overall, while this study is not particularly novel, it is technically sound, clearly written, and its data are appropriately analyzed and discussed. The findings provide some valuable guidance for selecting gene editing tools to study antibiotic resistance. I believe the work should be published once the points detailed below have been addressed.

Major points:

The claim that all three CRISPR-Cas systems achieved 100% efficiency is very strong, but the supporting data appears insufficient.

In lines 225-226, the description is unclear regarding how targets against the antibiotic resistance genes were selected and which specific targets were used in subsequent experiments.

In lines 238-239, the authors evaluated plasmid elimination efficiency using only five single colonies from a single experiment. This sample size seems too small to draw robust conclusions. Authors need to detail the number of biological replicates used in each experiment.

Many variables can impact CRISPR-Cas system targeting efficiency. The manuscript does not adequately define or address these factors.

Minor comments:

The manuscript requires careful English copy editing to address numerous grammatical and stylistic issues--there are too many issues to list here.

Line 51-52: The term "close position" is vague. The authors should specify what "close" refers to in this context.

Line 65: "Exhibit" should be corrected to "exhibits."

Strain names should be italicized (e.g., line 53, *Escherichia coli*; line 87, *Klebsiella pneumoniae*).

Lines 280-294: Correct the typos for "10⁴."

Lines 331-335: This section is unnecessarily redundant and should be streamlined.

Reviewer # 2 (comments for the author)

Authors have addressed an important issue of antimicrobial resistance. It's true that with growing resistance, antimicrobials are becoming ineffective. With CRISPR-Cas system, it has now been convenient to specifically target resistance and block its transmission. However, efficiency of each system is not well explained. Authors have assessed efficacy of three different types of CRISPR-Cas system. They have tried to explain use of CRISPR in eradicating resistance. In their work, they have clearly described each and every experiment properly and have also addressed each aim. The manuscript has been well written. Such work will pave newer techniques to address the problem of ever growing resistance.

Major comments:

Line no. 253-264: This study has targeted in elimination of carbapenem-resistant genes-KPC-2 and IMP-4. However, when assessing the susceptibility, none of the carbapenems were evaluated. Why is this so?

Minor comments:

Line no.54, 87: Italicize scientific names of organism.

Line no. 175: Do not italicize 5 α

Line no. 223: Add 'in' after 'three CRISPR systems'.

Line no. 304 & 309: Replace 're-coated' with 'spread'

Reviewers' comment-

Authors have addressed an important issue of antimicrobial resistance. It's true that with growing resistance, antimicrobials are becoming ineffective. With CRISPR-Cas system, it has now been convenient to specifically target resistance and block its transmission. However, efficiency of each system is not well explained. Authors have assessed efficacy of three different types of CRISPR-Cas system. They have tried to explain use of CRISPR in eradicating resistance. In their work, they have clearly described each and every experiment properly and have also addressed each aim. The manuscript has been well written. Such work will pave newer techniques to address the problem of ever growing resistance.

Major comments:

Line no. 253-264: This study has targeted in elimination of carbapenem-resistant genes-KPC-2 and IMP-4. However, when assessing the susceptibility, none of the carbapenems were evaluated. Why is this so?

Minor comments:

Line no.54, 87: Italicize scientific names of organism.

Line no. 175: Do not italicize 5α

Line no. 223: Add 'in' after '**three CRISPR systems**'.

Line no. 304 & 309: Replace 're-coated' with 'spread'

Microbiology spectrum

Ref: Spectrum02572-24

Comparison of CRISPR-Cas9, CRISPR-Cas12f1 and CRISPR-Cas3 in eradicating
resistance genes KPC-2 and IMP-4

Response Letter to Reviewers' Comments

Dear reviewer

We sincerely appreciate your valuable comments and professional suggestions, which have significantly enhanced the academic rigor of our manuscript. In response to your feedback, we have carefully revised the manuscript accordingly. We believe these modifications have further improved the quality of our work. Below, we provide a detailed summary of the changes made:

Reviewer #1

The manuscript by Huang et al. compared the efficiency of three CRISPR-Cas systems-CRISPR-Cas9, CRISPR-Cas12f1, and CRISPR-Cas3-in targeting antibiotic resistance genes KPC-2 and IMP-4 in *E. coli*. The authors demonstrated that all three systems achieved 100% eradication efficiency. Furthermore, they identified CRISPR-Cas3 as particularly effective in assisting antibiotics to kill drug-resistant bacteria.

Overall, while this study is not particularly novel, it is technically sound, clearly written, and its data are appropriately analyzed and discussed. The findings provide some valuable guidance for selecting gene editing tools to study antibiotic resistance. I believe the work should be published once the points detailed below have been addressed.

Major points:

The claim that all three CRISPR-Cas systems achieved 100% efficiency is very strong, but the supporting data appears insufficient.

Question1 : In lines 225-226. the description is unclear regarding how targets against

the antibiotic resistance genes were selected and which specific targets were used in subsequent experiments.

Answer1 : We sincerely appreciate this valuable and insightful comment. We have thoroughly reviewed the feedback and have implemented the necessary corrections. For each CRISPR system, three targets were designed. PCR analysis of the colonies demonstrated that each target successfully removed the resistance genes (Figures S2). Subsequently, target sites within the regions 542-576 bp of the KPC-2 gene and 213-248 bp of the IMP-4 gene were selected to compare their clearance efficiencies. (Figure S3).

(See line no. 243-251 of clean version)

Question2 : In lines 238-239, the authors evaluated plasmid elimination efficiency using only five single colonies from a single experiment. This sample size seems too small to draw robust conclusions. Authors need to detail the number of biological replicates used in each experiment.

Answer2 : We fully agree with this insightful suggestion. To enhance the reproducibility of our results, we increased the number of single colonies analyzed to 20, as depicted in Figure S2.

(See line no. 243-245 of clean version)

Question3 : Many variables can impact CRISPR-Cas system targeting efficiency. The manuscript does not adequately define or address these factors.

Answer3 : We totally agree with the reviewer that we need to define or address more targeting efficiencies that affect CRISPR-Cas systems. We have added relevant factors affecting the efficiency of CRISPR systems in the discussion section.

(See line no. 396-408 of clean version)

Minor comments:

The manuscript requires careful English copy editing to address numerous grammatical and stylistic issues--there are too many issues to list here.

Question4 : Line 51-52: The term "close position" is vague. The authors should

specify what "close" refers to in this context.

Answer4 : Thank you for your suggestion. The term "close position" refers to the design of target sites for the three CRISPR systems within the regions 542-576 bp of the KPC-2 gene and 213-248 bp of the IMP-4 gene, respectively. We have revised our description of the "close position" accordingly.

(See line no. 51-53,112-113 and 144-145 of clean version)

Question5 : Line 65: "Exhibit" should be corrected to "exhibits."

Answer5 : We sincerely appreciate your meticulous review and apologize for the oversight. In accordance with the reviewer's suggestion, we have revised "Exhibit" to "exhibits."

(See line no. 66 of clean version)

Question6 : Strain names should be italicized (e.g., line 53, *Escherichia coli*; line 87, *Klebsiella pneumoniae*).

Answer6 : We sincerely apologize for the oversight in our manuscript. We greatly appreciate your reminder and have now italicized the strain names as required.

(See line no. 54, 57, 60, 88, 126, 132, 170, 173, 236-237, 347, 429, 484, 498 and 504 of clean version)

Question7 : Lines 280-294: Correct the typos for " 10^4 ."

Answer7 : We appreciate your valuable suggestions and have revised the manuscript accordingly.

(See line no. 283, 285, 287, 292, 294 and 296 of clean version)

Question8 : Lines 331-335: This section is unnecessarily redundant and should be streamlined.

Answer9 : We appreciate your constructive feedback and have streamlined this part by removing the relevant sentences.

(See line no. 333-334 of clean version)

Reviewer # 2

Authors have addressed an important issue of antimicrobial resistance. It's true that with growing resistance, antimicrobials are becoming ineffective. With CRISPR-Cas system, it has now been convenient to specifically target resistance and block its transmission. However, efficiency of each system is not well explained. Authors have assessed efficacy of three different types of CRISPR-Cas system. They have tried to explain use of CRISPR in eradicating resistance. In their work, they have clearly described each and every experiment properly and have also addressed each aim. The manuscript has been well written. Such work will pave newer techniques to address the problem of ever-growing resistance.

Major comments:

Question1 : Line no. 253-264: This study has targeted in elimination of carbapenem-resistant genes-KPC-2 and IMP-4. However, when assessing the susceptibility, none of the carbapenems were evaluated. Why is this so?

Answer1 : We greatly appreciate the valuable feedback. Given that the drug resistance gene sequence was synthesized directly and the carbapenem resistance gene in the reconstructed plasmid was not fully expressed, whereas ampicillin resistance can be stably expressed on the recombinant plasmid, we opted to evaluate ampicillin resistance instead of carbapenem resistance.

Minor comments:

Question2 : Line no.54, 87: Italicize scientific names of organism.

Answer2 : We sincerely appreciate your meticulous review and apologize for the oversight. In accordance with the reviewer's suggestion, we have italicized the scientific names of organisms throughout the manuscript.

(See line no. 54, 57, 60, 88, 126, 132, 170, 173, 236-237, 347, 429, 484, 498 and 504 of clean version)

Question3 : Line no. 175: Do not italicize 5α

Answer3 : We sincerely apologize for the oversight in our manuscript. We have modified the format of 5α .

(See line no. 127, 170, 173, 281, 283, 286, 290, 292, 295, 429, 445 and 446 of clean version)

Question4 : Line no. 223: Add 'in' after 'three CRISPR systems'.

Answer4 : Thank you for pointing out this problem in manuscript. We have added 'in' after 'three CRISPR systems'.

(See line no. 220 of clean version)

Question5 : Line no. 304 & 309: Replace 're-coated' with 'spread'

Answer4 : We gratefully thanks for the precious time the reviewer spent making constructive remarks, we have replaced 're-coated' with 'spread'.

(See line no. 304 and 312 of clean version)

We sincerely appreciate your time and attention. We look forward to your response.

Yours sincerely,

Hongbin Song

19 January 2025

Department of Infectious Disease Prevention and Control,

Chinese People's Liberation Army Center for Disease Control and Prevention,

Beijing, 100071, China

Re: Spectrum02572-24R1 (Comparison of CRISPR-Cas9, CRISPR-Cas12f1 and CRISPR-Cas3 in eradicating resistance genes KPC-2 and IMP-4)

Dear Prof. Hongbin Song:

Your manuscript has been accepted, and I am forwarding it to the ASM production staff for publication. Your paper will first be checked to make sure all elements meet the technical requirements. ASM staff will contact you if anything needs to be revised before copyediting and production can begin. Otherwise, you will be notified when your proofs are ready to be viewed.

Sincerely,
Silvia Cardona
Editor
Microbiology Spectrum

Reviewer #1 (Comments for the Author):

This is the revision of the manuscript I reviewed. The authors have made considerable efforts to address the reviewers' comments and improve the quality of their work. I have no additional comments.